# Post-Fatigue Fracture Resistance of Lithium Disilicate and Polymer-Infiltrated Ceramic Network Indirect Restorations over Endodontically-Treated Molars with Different Preparation Designs: An In-Vitro Study

**DOI:** 10.3390/polym14235084

**Published:** 2022-11-23

**Authors:** Allegra Comba, Andrea Baldi, Massimo Carossa, Riccardo Michelotto Tempesta, Eric Garino, Xhuliano Llubani, Davide Rozzi, Julius Mikonis, Gaetano Paolone, Nicola Scotti

**Affiliations:** 1Department of Surgical Sciences, C.I.R. Dental School, University of Turin, Via Nizza 230, 10126 Turin, Italy; 2Private Practice, 54073 Kaunas, Lithuania; 3Department of Dentistry, IRCCS San Raffaele Hospital and Dental School, Vita Salute University, 20132 Milan, Italy

**Keywords:** polymer-infiltrated ceramic network, lithium disilicate, indirect restorations, preparation design, fatigue test, endodontically-treated teeth

## Abstract

The aim of the present study was to evaluate the fatigue to cyclic and static resistance of indirect restorations with different preparation designs made either of lithium disilicate (LS) or polymer-infiltrated ceramic network (PICN). Eighty-four (*n* = 84) molars were chosen, endodontically treated, and prepared with standardized MOD cavities. The molars were randomly divided into 6 study groups (*n* = 14) taking into account the “preparation design’’ (occlusal veneer with 1.2 mm occlusal thickness; overlay with 1.6 mm occlusal thickness; adhesive crown with 2 mm occlusal thickness) and the “CAD/CAM material’’ (E-max CAD, Ivoclar vivadent; Vita Enamic, Vita). A fatigue test was conducted with a chewing simulator set at 50 N for 1,500,000 cycles. Fracture resistance was assessed using a universal testing machine with a 6 mm diameter steel sphere applied to the specimens at a constant speed of 1 mm/min. A SEM analysis before the fracture test was performed to visually analyze the tooth-restoration margins. A statistical analysis was performed with a two-way ANOVA and a post-hoc pairwise comparison was performed using the Tukey test. The two-way ANOVA test showed that both the preparation design factor (*p* = 0.0429) and the CAD/CAM material factor (*p* = 0.0002) had a significant influence on the fracture resistance of the adhesive indirect restorations. The interaction between the two variables did not show any significance (*p* = 0.8218). The occlusal veneer had a lower fracture resistance than the adhesive crown (*p* = 0.042) but not lower than the overlay preparation (*p* = 0.095). LS was more resistant than PICN (*p* = 0.002). In conclusion, in the case of endodontically treated teeth, overlay preparation seems to be a valid alternative to the traditional full crown preparation, while occlusal veneers should be avoided in restoring non-vital molars with a high loss of residual tooth structure. LS material is more resistant compared to PICN.

## 1. Introduction

Indirect bonded restorations—such as inlays, onlays, and overlays—represent a routine treatment option to restore vital teeth when a destructive carious lesion or a cuspal fracture occurs [1,2]. Recently, indirect bonded restorations have also been proposed to rehabilitate endodontically-treated teeth (ETT), as an alternative to the traditional full crown coverages [3,4,5]. Endodontic treatments have reached a high level of predictability, allowing the maintenance nowadays of teeth that would otherwise have had to be extracted [6,7,8]. However, ETT differ from vital teeth due to the large loss of structure that lead to several mechanical alterations, such as greater fragility [9]. Moreover, changes in the structural architecture of the dentin and the loss of water lead to greater brittleness of the ETT compared to healthy teeth [10]. As a result, they are prone to greater cusp deflection during functioning and, consequently, are usually more prone to fractures [11]. In the case of ETT, the full crown still represents the most documented treatment option to restore function and aesthetic, to provide endodontic seal, as well as to increase the longevity of the treated tooth [3,4,12]. However, it also represents the most invasive treatment regarding the removal of tooth structure [13,14]. For this reason, indirect adhesive restorations, which represent a less invasive treatment with respect to the sound tissue [13,14], has started to be introduced as an alternative treatment option for ETT. Indeed, indirect restorations are reported to represent a more conservative treatment option and, simultaneously, are equally effective in terms of mechanical, functional, and aesthetic properties [15,16].

Materials of choice for indirect restorations have shifted during the years from metal and gold alloy to the more recent metal-free materials [17]. The majority of these materials can be manufactured with computer-aided design/computer aided manufacturing (CAD/CAM) technologies. In fact, CAD/CAM processes have today gained a predominant role in manufacturing metal-free indirect restorations [18].

Among different materials, lithium disilicate (LS) and polymer-infiltrated ceramic network (PICN) are two of the most commonly used materials for indirect bonded restorations. LS is a reinforced glass ceramic material that has been demonstrated to have excellent strength and optimal aesthetic properties [19]. Different studies have investigated its properties in both in vitro and in vivo scenarios [19,20]. Malchiodi et al. [21] followed 43 LS onlays over a mean observational period of 32 months, finding a survival rate of 97.7%. Ghert et al. [22] analyzed 104 LS single adhesive crowns over an observational period of 9 years. The authors’ findings showed a survival rate of 97.4% after 5 years and 94.8% after 8 years. This result is in agreement with the study of Simeone and Gracis [23] who found a survival rate of 98.2% following 275 single LS adhesive crowns over an observational period of 11 years. Consequently, it can be assumed that LS represents an excellent material option for indirect restorations.

On the other hand, PICNs represent a combination of ceramic and resin composites, consisting of the structure of a sintered ceramics matrix (86% in weight) infiltrated with a polymer matrix (14% in weight). This material exhibits a low elastic modulus, great mechanical strength, and was proven to have less strength degradation than most ceramics [24]. Moreover, it was recently demonstrated how PICN materials provide advantages in stress distribution by homogeneously distributing the stress into the cement surface [24]. On the contrary, LS was seen to distribute the stress in localized points, mainly located in the tooth structure [25].

Among other parameters, fatigue resistance is considered one the primary parameters to predicting the future outcomes of an indirect restoration. While data about the fatigue resistance of LS are well reported [18], few articles are currently available on the fatigue resistance of PICNs. Nonetheless, the comparison of LS and PICNs data from the literature is controversial [26,27,28,29]. Aboushelib et al. [26] showed how the internal structure of PICN materials favors their fatigue resistance in comparison to ceramic materials. Consistencies were shown by Swain et al. [27] who found better outcomes for PICNs compared to LS after being exposed to mechanical cycling. However, results from the study of Homai et al. [28] disagreed with the above-cited articles. In their study, the authors tested the fatigue resistance of PICN and LS crowns and found no survival at the end of the test for PICN crowns, while 70% of LS crowns survived. Consequently, discrepancies are present in the literature and research on the topic remains open.

Thus, the aim of this in vitro study was to evaluate the fracture resistance of endodontically-treated molars restored with three different preparation designs (adhesive crown, overlay, or occlusal veneer) combined with two different CAD/CAM monolithic materials (LS or PICNs). The null hypotheses tested were that fatigue resistance is not influenced by (1) the preparation design, nor by (2) the material employed.

## 2. Materials and Methods

### 2.1. Study Design

Table 1 provides a general description of the CAD/CAM materials used in the current study, their manufacturers, and their composition. Sample size calculation was performed with G*Power 3.1 (Kiel University, Kiel, Germany) to set the study power higher than 90% considering an alpha error = 0.05.

Specimens were randomly (https://randomizer.org accessed on 20 October 2021) distributed among 6 study groups (*n* = 14), taking into account:(1)Tooth preparation designs at three levels: adhesive crown, overlay, and occlusal veneer, there being one instance where only one bulk-fill composite resin (Voco, Cuxhaven, Germany) was used for the build-up core;(2)“CAD/CAM blocks” at two levels: after core build-up, two different CAD/CAM restorative materials were tested: a PICN (GrandioBlocks, Voco, Cuxhaven, Germany) and an LS (E-max CAD, HT A2/C14, Ivoclar Vivadent).

### 2.2. Specimen Selection

Eighty-four (*n* = 84) human upper maxillary molars with mature apices that had been extracted for periodontal compromise during the previous four months were chosen and kept at room temperature in distilled water. Age of the donors ranged from 41 to 67 (mean 53 ± 13) years old. Inclusion criteria considered were: sound teeth, comparable roots (length > 12 mm), crown sizes (10 ± 2 mm mesio-distal, 10 ± 2 mm bucco-oral), and the absence of cracks or demineralization upon ocular inspection with light trans-illumination and magnification. For surface debridement, hand instrumentation comprising ultrasonic scaling and polishing was used. Following washing processes, specimens were kept in distilled water at room temperature for at least 72 h [30]. All samples were acquired with informed and signed permission in the Department of Cariology and Operative Dentistry at the C.I.R. Dental School of the University of Turin.

### 2.3. Endodontic Treatment

All specimens underwent endodontic treatment, which was carried out by the same skilled operator with the working length set at 1 mm short of the apparent apical foramen. Proglider and ProTaper Next X1-X2 (Dentsply Maillefer, Ballaigues, Switzerland) were used to perform canal shaping. An amount of 5% NaOCl (Niclor 5, Ogna, Muggi, Italy) and 10% EDTA (Tubuliclean, Ogna, Milan, Italy) were alternately used for irrigation. Then, utilizing Down Pack (Hu-Friedy, Chicago, IL, USA) and an endodontic sealer (Pulp Canal Sealer EWT, Kerr, Orange, CA, USA), canals were obturated with gutta-percha points (GuttaPercha Points Medium, Inline, B.M. DentaleSas, Turin, Italy). Backfilling with gutta-percha (Obtura III system, Analytic Technologies, Redmond, WA, USA) was then performed [29].

### 2.4. Specimen Preparation

One skilled operator prepared standardized MOD cavities. The mesial and distal cervical margins were positioned at 1 mm coronally to the CEJ, and the residual wall thickness of the buccal and oral cusps was fixed at the height of the contour to 1.5 ± 0.2 mm. To prepare the cavity, a high-speed headpiece (Kavo Dental GmbH, Biberach, Germany) was utilized with cylindrical diamond burs (835KR, Komet, Schaumburg, IL, USA) that were heavily cooled by air and water. To remove non-sustained enamel, an Arkansas point (FG 645, Komet, Schaumburg, IL, USA) was used to round and smooth internal edges. The cavities underwent adhesion procedures as follows: 30 s of selective enamel etching with 35% phosphoric acid (K-etchant, Kuraray Noritake Dental, Tokyo, Japan), followed by 30 s of rinsing and 30 s of air drying. Following the manufacturer’s instructions, a universal adhesive system (Futurabond U, Voco, Cuxhaven, Germany) was applied. It was then light-cured for 20 s using an LED light-curing device at 1000 mW/cm2 (Cefalux 2, Voco, Cuxhaven, Germany). With a bulk-fill material, the MOD cavities were then horizontally gradually restored (Grandioso X-Tra, Voco, Cuxhaven, Grmany). The same curing LED lamp was used to light-cure each layer, maximally 3 mm thick, for 30 s.

Following this, all specimens were split into 3 groups (*n* = 28 each) based on the indirect adhesive preparation carried out:Group 1:Occlusal veneer. A cylindrical bur (6836 KR 014, Komet) was used to perform a standardized 1.2 mm occlusal reduction following the occlusal anatomy. A conical bur with a flat point (H15809, Komet) was used to create 1 mm deep mesial and distal interproximal boxes. Finally, a football-shaped bur (8368 L, Komet) was used to bevel the occlusal margins (8368 L, Komet), and an Arkansas tip (661, Komet) and a rubber point (9436 M, Komet) were used to round all corners and finish the preparation.Group 2:Overlay. A 1.6 mm occlusal reduction was performed using a cylindrical bur (6836 KR 014, Komet). A round shoulder of 1.5 mm depth, placed in the middle third of the clinical crown, and interproximal boxes, placed 1 mm above the CEJ, were then performed in all specimens with a conical bur with a flat point (H15809, Komet). An Arkansas tip (661, Komet) and a rubber point (9436 M, Komet) were used to round all corners.Group 3:Adhesive crown. A standardized 2 mm occlusal reduction was performed with a chamfer margin 1  ±  0.5 mm above CEJ. Chamfer burs (6881 014, Komet; 8881 014, Komet) were used to perform initial and finishing preparations. Then, an Arkansas tip (661, Komet) and a rubber point (9436 M, Komet) were used to round all corners.

Then, prepared specimens were scanned with an intraoral optical camera (Omnicam 2.0, Dentsply, Sirona, Konstanz, Germany) and the preparation margins were identified with CAD/CAM Chairside software (Cerec 4.5.2). Indirect adhesive restorations were digitally designed with a similar occlusal anatomy and an occlusal thickness according to the different preparation designs (1.2 mm for occlusal veneers, 1.6 mm for overlays, and 2 mm for adhesive crowns). Specimens were then divided into 2 subgroups (*n* = 42) according to the CAD/CAM material selected to mill the restoration:Subgroup A: PICN (Vita Enamic, Vita), shade 2M1-HT.Subgroup B: LS (E-max CAD, Ivoclar), shade A2 HT.

Both materials were milled in extra-fine mode with a 4-axis milling machine (MXCL) in wet conditions. E-max was then sintered following the manufacturer instructions. All restorations were finally polished with rotation silicon points and brushes.

### 2.5. Luting Procedure

Following the manufacturer’s recommendations, each indirect restoration was luted using a universal adhesive (Futurabond U, Voco, Cuxhaven, Germany) and a dual-curing cement (Bifix QM, Voco, Cuxhaven, Germany). Prior to silane application, restoration surfaces were etched with 5% hydrofluoric acid for 20 s (LS) and 60 s (PICN), and then cleaned with an ultrasonic bath. After placing the restoration and using brushes to remove any excess cement, the same LED curing light was used for a 60 s light-cure on each side (Valo, Ultradent). After covering the samples with clear air barrier gel, a final polymerization lasting 20 s per side was carried out. Using silicone cups and diamond burs, the surface was finished and polished until smooth [30].

### 2.6. Cyclical Intermittent Loading

Using a CS-4.4 chewing simulator, samples were subjected to cyclic intermittent loading in distilled water (SD Mechatronik, Feldkirchen-Westerham, Germany). A 0.3 mm thick silicon layer was used to surround the teeth roots to simulate a periodontal ligament. In accordance with other studies on fatigue testing, 50 N of force was delivered using 6 mm diameter steatite balls as antagonists at the following settings: 1,500,000 cycles, a frequency of 1 Hz, a speed of 16 mm/s, and a sliding distance of 2 mm along the buccal triangle crest [30].

Events which compromised the restoration or the supporting tissue integrity during the cycling fatigue test were recorded.

### 2.7. Fracture Resistance

A static fracture resistance test was performed on specimens that survived the cyclic fatigue test using a universal testing machine (Instron, Canton, MA, USA). The machine was equipped with a crosshead made of steel with a diameter of 6 mm that was welded to a tapered shaft. The crosshead was applied to the specimens at a constant speed of 1 mm/min and parallel to the long axis of the tooth. In Newton, maximum fracture loads were noted for statistical purposes. The failure modalities of fractured specimens were evaluated, and catastrophic fractures (non-repairable, below the CEJ) and non-catastrophic fractures (reparable, above the CEJ) were identified. The classification was based on a three-examiner consensus [30].

### 2.8. Scanning Electron Microscopy (SEM) Analysis

In order to show the different surface characteristics of the tested materials related to different preparation designs before and after chewing simulation, both baseline and after-completion of Cyclical Intermittent Loading specimen impressions were made (Flecitiem Extra-light, Kulzer, Hanau, Germany), and replicas were created (Alpha Die, Schütz Dental, Rosbach, Germany). The finished replicas were placed on aluminum stubs, gold was sputter-coated on them, and they were then analyzed under SEM (Phenom, FEI, Amsterdam, Netherlands) at a magnification of 500×. The marginal quality of interfaces was assessed using SEM analysis [31].

### 2.9. Statistical Analysis Description

A Shapiro-Wilk test revealed that the data were normally distributed. A two-way analysis of variance (ANOVA) test was carried out to investigate the effects of the study factors (preparation design and CAD/CAM material) and their interactions on fracture resistance. The Tukey test was used to perform a post hoc pairwise comparison. The same software was used for all statistical analyses (STATA, ver. 12.0, StataCorp, College Station, TX, USA) and differences were deemed significant at *p* < 0.05.

## 3. Results

Events observed during cyclical intermittent loading are reported in Table 2. Any debonding was observed in occlusal veneers and overlay groups, which showed some partial fractures that never extended over the remaining tooth structure. On the other hand, the adhesive crowns showed one core fracture.

Table 3 lists the mean fracture resistance to static load, expressed in N, for the various groups.

A two-way ANOVA test revealed that both the preparation design factor (*p* = 0.0429) and the CAD/CAM material factor (*p* = 0.0002) had a significant influence on the fracture resistance of the adhesive indirect restorations over endodontically-treated molars. The post-hoc Tukey test revealed that, regarding the preparation design, the occlusal veneer had a lower fracture resistance than the adhesive crown (*p* = 0.042) but not lower than the overlay preparation (*p* = 0.095). Regarding the CAD/CAM material factor, LS was more resistant than PICN (*p* = 0.002). The interaction between the two variables did not show any significance (*p* = 0.8218).

Failure modalities are reported in Table 4.

The wear pattern of the tested materials was visible in representative SEM images (Figure 1, Figure 2 and Figure 3).

## 4. Discussion

The aim of the present in vitro study was to evaluate the cyclic and static fatigue resistance of endodontically-treated molars restored with three different preparation designs (adhesive crown, overlay, or occlusal veneer) combined with two different CAD/CAM monolithic materials (LS or PICN). Based on the results of the present study, the occlusal veneer showed a statistically significant lower fracture resistance compared to adhesive crowns (*p* = 0.0429). Consequently, the first null hypothesis was rejected. On the other hand, the maximum fracture resistance was observed with the overlay preparation design (2059.5 ± 308.0 N). However, no significant difference was seen between the adhesive crown and overlay, nor between the overlay and occlusal veneer.

For many years, the full crown, which represents the most invasive preparation design according to the studies of Edelhoff and Sorensen [13,14], was considered the only treatment option to restore ETT. Undoubtedly, ETT are disadvantaged in terms of mechanical resistance [32,33] and substrate quality [34,35] compared to vital teeth. In fact, a recent systematic review and meta-analysis by Dioguardi et al. [36] compared indirect bonded restoration on vital and ETT, showed how the risk of failure is higher for ETT. However, with the advances of adhesive procedures [37] and modern materials [38,39], indirect bonded restorations have started to be considered as an alternative treatment option for ETT [40]. Studies about preparation designs on ETT are useful for answering the question of whether minimally invasive treatment can be used as an alternative to the traditional full crown workflow. Data from the present study showed that the less invasive preparation design (occlusal veneer) exhibited less resistance to static fracture compared to the more invasive procedure (adhesive full-crown). On the other hand, an overlay was seen to be a valid alternative to an adhesive crown, since no statistical difference was highlighted. In regard to the occlusal veneer, the result of the present study is in agreement with the study of Frankenberger et al. [41] found how minimally invasive preparations were less successful in the rehabilitation of ETT compared to more invasive ones. It should be pointed out that the occlusal veneer is today indicated for the rehabilitation of worn dentition more than for the reinforcement of compromised clinical crowns, such as non-vital molars are. In regard to the other two preparation designs considered, the data of the present study are in agreement with Skouridou et al. [42] who found a similar fracture strength between a full crown and less invasive indirect restoration, such as overlays. Consistency is also demonstrated by the systematic review and meta-analysis of the in vivo study by Vagropoulou et al. [43] which found a comparable survival rate (<90%) between full crowns and overlays after a 5 years observational period. However, the present result is in disagreement with the study of Jurado et al. [44] which found higher fracture resistance for full crowns compared to overlays.

However, consideration should be given to the in vitro analysis performed when testing a restorative solution. A post-fatigue fracture resistance test allows one to compare the residual resistance to a static load on a material that has undergone a cyclic functional stress. However, several recent studies have shown that dynamic loading and artificial aging tests are a reliable method to providing clinically relevant information on the long-term stability, reliability, limitations, and lifetime predictions of monolithic restorations [26,45]. Therefore, the failures that occur during a cyclic chewing simulation test seem to be more indicative of the clinical behavior that a material, especially ceramic, may show when used for the restoration of a compromised tooth. In the present study, none of the minimally invasive solutions tested showed catastrophic failures during the cyclic fatigue test, while the adhesive crown experienced one core fracture.

Another issue which should be considered during fatigue testing is the marginal gap opening [46,47]. A combination of compressive and lateral forces on indirect adhesive restorations could lead forces to be transmitted to the adhesive interface and the supporting tooth towards the restorative material [30]. The sliding motion used in the fatigue simulation is intended to increase the lateral forces acting on the restorative material and cause the system to flex. Previous research indicated that wide adhesive interfaces in high-retentive solutions may have served as a cushion, better dissipating forces and preventing gap progression. [46,48]. In the present study, the adhesive crown showed a greater tendency to develop adhesive interface gaps than minimally invasive solutions, probably because of the smaller amount of sound enamel at the cervical area of a molar (Figure 1, Figure 2 and Figure 3). In this regard, scientific research remains focused on indirect bonded restorations, in contrast to the more proven full crown, due to the preservation of sound tissue they offer. Less retentive preparations, with the maximum conservation of sound structure, were also proven to reduce the polymerization shrinkage stress [49] and to positively influence the mechanical behavior of the restorations when compared to more retentive preparation designs [25]. Indeed, the preservation of the remaining tooth structure may be a key factor for the long-term prognosis of the ETT [4,50]. From a clinical standpoint, preparation design should be decided according to the residual tooth structure. In the case of an ETT, overlay preparation seems to be a valid alternative to the full crown, with the advantages of preserving the largest amount of sound structure and showing fewer destructive failures during dynamic loading [13,14].

On the other hand, in regard to the fatigue resistance of LS and PICN, a statistically significance difference was found between the two materials, with LS exhibiting the higher fracture resistance. Therefore, the second null hypothesis was also rejected. However, this result should be interpreted with caution. After the cycling intermittent load test, with 80 N repeated over 1,500,000 cycles simulating an oral environment over 5 years, no failures were observed for PICN while 1 core fracture was observed for LS. Later, specimens were submitted to a static increasing load until fracture, which is useful to test the absolute fracture resistance of a material. At the end of the test, PICN was seen to be less resistant compared to LS. Consequently, the data of the present study indicate that both materials seem to be valid alternatives when oral simulation loads are considered, while with the increasing load, LS was found to be preferable compared to PICN. The present result is in agreement with an extensive review on the topic by Facenda et al. [24] which concluded that PICN materials seem to show high fracture resistance when physiological loadings (up to 200 N) are applied. When the load increases, other materials, such as LS, seem to be favored.

Data about fracture resistance of PICNs are currently controversial. The present results partially agree with the study of Swain et al. [27] which found no failures of crowns made of PICNs after mechanical cycling and 6 minor cracks highlighted for LS crowns. Concerning PICNs, similar results were found by El Zhawi et al. [51], who found an excellent resistance to fatigue for the tested material. However, the studies of Swain [27] and El Zhawi [51] differ from the present article in methodology, since the crowns were cemented on composite resin dyes. On the other hand, different results were found by Homai et al. [28], who compared crowns made either with PICNs or LS on human premolars. The authors tested the fatigue resistance of the materials by performing an increasing mechanical chewing test (starting from 500 N, the load was increased by 100 N every 100,000 cycles until fracture). The study findings showed no survival at the end of the test for PICN crowns, while 70% of LS crowns survived. These controversial results, with apparently similar in vitro investigations, may be related to the various testing procedures used. From a clinical standpoint, PICN materials are easier to manage compared to LS, since they do not need to be crystalized, reducing the number of steps and the manufacturing time. Moreover, they may offer some advantages in terms of reparability linked to their composition in case small chippings occur.

However, based on the rapid evolution of novel CAD/CAM materials, further in vitro and in vivo studies are required to investigate the potential and limitations of these materials.

The limitations of the present study are inherent to the in vitro design, where only controlled variables are considered. No thermal stresses during the cyclic fatigue test were applied. Intra-oral temperature changes may influence the long-term outcome of indirect restoration since the different materials employed in the study present higher thermal contraction/expansion coefficients than tooth tissues. 

## 5. Conclusions

Within the limitations of the present in vitro study, based on the obtained results the following conclusions can be drawn: −Different preparation designs and CAD/CAM materials seem to influence the fatigue resistance of adhesive indirect restorations over ETT;−Overlay preparation seems to be a valid alternative to the traditional full crown, while occlusal veneers should be avoided in restoring non-vital molars with a high loss of residual tooth structure;−Both LS and PICN showed an optimal survival rate after a cycling intermittent load test;−Occlusal veneers and overlay never showed dramatic failures during dynamic loading, showing a high ability to protect the supporting tooth structure;−Regarding static fracture resistance, LS is more resistant than PICN.

## Figures and Tables

**Figure 1 polymers-14-05084-f001:**
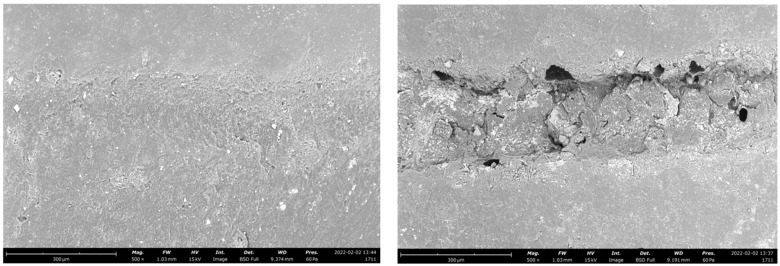
Representative SEM images of the adhesive interface of an adhesive crown before (**left**) and after (**right**) the chewing simulation test, restored with lithium disilicate.

**Figure 2 polymers-14-05084-f002:**
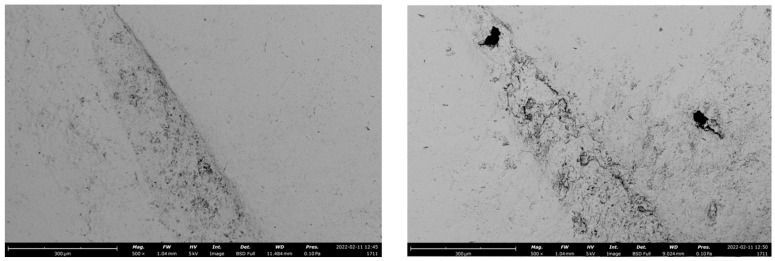
Representative SEM images of the adhesive interface of an adhesive crown before (**left**) and after (**right**) the chewing simulation test, restored with PICN.

**Figure 3 polymers-14-05084-f003:**
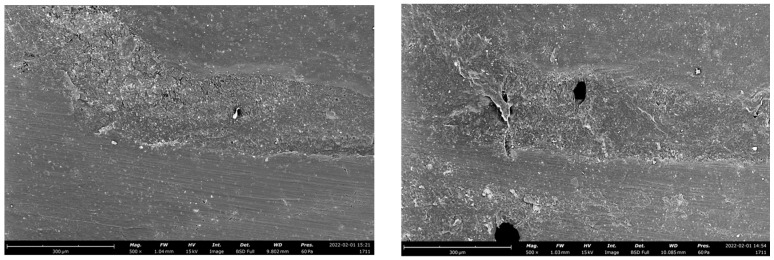
Representative SEM images of the adhesive interface of an occlusal veneer before (**left**) and after (**right**) the chewing simulation test, restored with PICN.

**Table 1 polymers-14-05084-t001:** General description, manufacturers, and composition of the CAD/CAM blocks employed in the study.

Material	General Description	Manufacturer	Composition
E-max CAD	Lithium disilicate	Ivoclar vivadent, Schaan, Liechtenstein	SiO_2_ 60.0–65.0%, K2 = 15.0–19.0%, Al2 = 3 6.0–10.5%, other oxides and pigments 0.0–8.0%
Vita Enamic	Polymer-infiltrated ceramic network	Vita, Bad Säckingen, Germany	Feldspar ceramic enriched with aluminum oxide (75% *v*/*v*), (wt 86%), UDMA, TEGDMA (14% wt 25% *v*/*v*)

**Table 2 polymers-14-05084-t002:** Events observed during cyclical intermittent loading.

	Occlusal Veneer		Overlay		Adhesive Crown	
	LS	PICN	LS	PICN	LS	PICN
**Events during cyclical intermittent loading**	1 chipping	3 chippings	none	2 chippings	1 debonding, 1 core fracture	2 debondings

**Table 3 polymers-14-05084-t003:** Mean fracture resistance ± standard deviation, expressed as Newton, for each group. Same superscript letters indicate no significant differences.

	Occlusal Veneer		Adhesive Crown		Overlay	
	PICN	LS	PICN	LS	PICN	LS
**Fracture resistance (N)**	1806.6 ± 270.1 ^ab^	2029.5 ± 295.3 ^a^	1555.4 ± 393.9 ^b^	1859.2 ± 232.0 ^ab^	1726.9 ± 301.1 ^ab^	2059.5 ± 308.0 ^a^

**Table 4 polymers-14-05084-t004:** Failure modalities of the observed specimens.

Fracture Pattern	Occlusal Veneer		Overlay		Adhesive Crown	
	LS	PICN	LS	PICN	LS	PICN
**Non-catastrophic**	75%	71.4%	71%	65.5%	50%	72.7%
**Catastrophic**	25%	28.6%	28%	34.5%	50%	27.3%

## Data Availability

The datasets generated and/or analyzed during the current study are available from the corresponding author on reasonable request.

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
