# Peer review of "Post-Fatigue Fracture Resistance of Lithium Disilicate and Polymer-Infiltrated Ceramic Network Indirect Restorations over Endodontically-Treated Molars with Different Preparation Designs: An In-Vitro Study"

_polymers, 2022, doi:10.3390/polym14235084_

Round 1

Reviewer 1 Report

This is a well-performed and well-documented study on the mechanical properties of two contemporary dental materials for indirect restorations. The Introduction section is clear and concise, providing just enough background information even for a non-dentist reader to understand the rationale for the study. The Materials and methods section is sufficiently detailed with a precise description of each procedure performed. The results are clearly presented and the discussion appropriately addresses all implications of the current study. Please consider a few minor comments for possible improvements to this otherwise excellent study:

Abstract: “In conclusion, in case of endodontically treated teeth, overlay preparation seems to be a valid alternative to the traditional full crown preparation, while minimal invasive preparations should be avoided.” – This is a reasonable conclusion but maybe avoid saying that “minimal invasive preparations should be avoided” because minimal invasive treatment is really never an option once teeth become endodontically involved. The term “minimally invasive” has a specific meaning in dentistry which does not the situation simulated in the present study.

Also abstract: Maybe mention there was no interaction in the two-way ANOVA. Some readers might be interested to get that information early.

Introduction: “The null hypotheses tested were that fatigue resistance is not influenced by the (1) preparation design, as well as by the (1) material employed.” – This numbering should be (1), (2)

Sample size: how did you arrive at exactly n=14? Was a power analysis performed; what was the desired power?

Could you specify the age of the tooth donors? This was supposedly an older age group, as the teeth were extracted due to periodontal indications. Please at least specify the age range.

Table 3: High data variability is expected in mechanical tests but for the presented data it is unnecessary to report them to two decimals. The second decimal is meaningless given the variability. This could be reduced to one decimal.

Author Response

Abstract: “In conclusion, in case of endodontically treated teeth, overlay preparation seems to be a valid alternative to the traditional full crown preparation, while minimal invasive preparations should be avoided.” – This is a reasonable conclusion but maybe avoid saying that “minimal invasive preparations should be avoided” because minimal invasive treatment is really never an option once teeth become endodontically involved. The term “minimally invasive” has a specific meaning in dentistry which does not the situation simulated in the present study.

Authors response: We thank the Reviewer for the comments. The abstract was modified according to it.

Also abstract: Maybe mention there was no interaction in the two-way ANOVA. Some readers might be interested to get that information early.

Authors response: We thank the reviewer for the comment. The interaction result between the two variables was added in the abstract. ‘‘Two-way ANOVA and Post-hoc pairwise comparison was performed using the Tukey test. Two-way ANOVA test showed that both the preparation design factor (p = 0.0429) and the CAD/CAM material factor (p = 0.0002) had a significant influence on the fracture resistance of the adhesive indirect restorations. The interaction between the two variables did not show any significance (p = 0.8218)…..’’

Introduction: “The null hypotheses tested were that fatigue resistance is not influenced by the (1) preparation design, as well as by the (1) material employed.” – This numbering should be (1), (2)

Authors response: We thank the Reviewer for the comments. The text was modified according to it.

Sample size: how did you arrive at exactly n=14? Was a power analysis performed; what was the desired power?

Authors response: We thank the Reviewer for the comment. Sample size calculation was provided in the materials and methods section.

Could you specify the age of the tooth donors? This was supposedly an older age group, as the teeth were extracted due to periodontal indications. Please at least specify the age range.

            Authors response: The range was 41-70 (mean 53 ± 13) years old. It was added in the text.

Table 3: High data variability is expected in mechanical tests but for the presented data it is unnecessary to report them to two decimals. The second decimal is meaningless given the variability. This could be reduced to one decimal.

           Authors response: We thank the Reviewer for the comment. The second decimal was removed from the data in Table 3.

Reviewer 2 Report

The paper entitled "Fatigue resistance of Lithium Disilicate and Polymer-Infiltrated Ceramic Network indirect restorations over Endodontically-
Treated Molars with different preparation designs: an in-vitro
study" is a study case report on the utilization of lithium disilicate and polymer-infiltrated ceramic network in dental restoration. The paper is for the purpose of the special issue.

I appreciate the composition of this paper, the analysis of the data by using statistics and the interpretation of the results according to the statistical data. But it is preferable that in Introduction Section the authors to highlight more the use of the polymer-infiltrated ceramic network in such applications. The authors should add some comments regarding the cited papers [24-27], not only to mention that it is controversial. From this point they must show the motivation to study these materials from their perspective.

In Discussion section the authors should add some literature comparisons.

In Conclusions Section they should mention the optimal design corresponding to the higher fracture resistance. Also, what are the benefits of PICN for these studies? The conclusion clearly shows the major contribution of LS.

Based on these comments, I recommend the publication of this paper after Minor revision.

Author Response

Reviewer 2

The paper entitled "Fatigue resistance of Lithium Disilicate and Polymer-Infiltrated Ceramic Network indirect restorations over Endodontically-Treated Molars with different preparation designs: an in-vitro
study" is a study case report on the utilization of lithium disilicate and polymer-infiltrated ceramic network in dental restoration. The paper is for the purpose of the special issue. I appreciate the composition of this paper, the analysis of the data by using statistics and the interpretation of the results according to the statistical data.

          Authors response: The Authors thank the Reviewer for the kind comments about the study and for the time spent reviewing it.

But it is preferable that in Introduction Section the authors to highlight more the use of the polymer-infiltrated ceramic network in such applications. The authors should add some comments regarding the cited papers [24-27], not only to mention that it is controversial. From this point they must show the motivation to study these materials from their perspective.

                Authors response:  we thank the Reviewer for the comment. Comments about the cited articles, as well as the different results obtained by the different Authors leading to the aim of the study, were added in the introduction.

In Discussion section the authors should add some literature comparisons.

               Authors response: we thank the Reviewer for the comment. Additional literature comparision was added in the discussion   

In Conclusions Section they should mention the optimal design corresponding to the higher fracture resistance.

             Authors response: We thank the Reviewer for the comment.I was added in the text.

Also, what are the benefits of PICN for these studies? The conclusion clearly shows the major contribution of LS.

            Authors response: We thank the Reviewer for the comment. Clinical advantages of PICN compare to LS were added in the discussion.

Based on these comments, I recommend the publication of this paper after Minor revision.

The Authors thank the Reviewer.

Reviewer 3 Report

The aim of the present study was to evaluate the fatigue to cyclic and static resistance of indirect restorations with different preparation designs made either of lithium disilicate or polymer-infiltrated ceramic network. This subject is interesting, however the text should be improved:

Title:

Most of your data was obtained after fatigue. Adjust your title and aim according to the second paragraph of your discussion: “post-fatigue fracture resistance” is more adequate.

Abstract:

Describe the restoration thickness;

What was the average fracture load per group?

Introduction:

Please describe how these different materials can affect the stress distribution in the restoration.

Check the Mechanical Behavior of Different Restorative Materials and Onlay Preparation Designs in Endodontically Treated Molars.

Methods:

Provide the sample size calculation;

Did you randomly distributed the teeth between the groups?

How was the finishing procedure? Glazing? Polishing? It was the same between both materials?

Information about surface etching is missing. PICN and LD present different silica content and therefore should be properly etched. Please clarify.

Why 50 N in the chewing load simulator? This is a very low load for posterior tooth.

How the “events were recorded” during fatigue? In steps? After a specific number of cycles? Automatically by the fatigue machine software?

What was the aim of topic 2.8? It is not clear.

The magnification described in the text and in the figures are different.

Provide the normality test to justify your parametric statistic.

Describe the Tukey test in table 3.

SEM figures are not clear. What should the reader see there? what region of the restoration are those?

Discussion:

Discussion should be improved with polymerization shrinkage stress between different preparation. Check the reference previously suggested and improve your text.

What is the effect of different elastic modulus in the set?

Discuss how handle both materials are, since one you need to crystalize before its use and the other is already fully-finished in the block.

How the simulation or not of periodontal ligament could affect your results?

Author Response

The aim of the present study was to evaluate the fatigue to cyclic and static resistance of indirect restorations with different preparation designs made either of lithium disilicate or polymer-infiltrated ceramic network. This subject is interesting, however the text should be improved:

Title:

Most of your data was obtained after fatigue. Adjust your title and aim according to the second paragraph of your discussion: “post-fatigue fracture resistance” is more adequate.

               Authors response: we thank the Reviewer for the comment. The tile was modified accordingly.

Abstract:

Describe the restoration thickness;

              Authors response: we thank the Reviewer for the comment. Occlusal thickness of each preparation design was added in ther Abstract.

What was the average fracture load per group?

            Authors response:  Considering the high number of words that need to be added per each group (Preparation design + material + average fracture load) and considering the limit of 200 words in the abstract section, the Authors preferred to maintain that average fracture load per group described in the main text and summarized in a Table (Table 3). The main results are highlighted in the abstract.

Introduction:

Please describe how these different materials can affect the stress distribution in the restoration.

Check the Mechanical Behavior of Different Restorative Materials and Onlay Preparation Designs in Endodontically Treated Molars. 

              Authors response: The Authors thank the Reviewer for pointing out this interesting topic. It was added in the introduction and the article was cited to support the topic.

Methods:

Provide the sample size calculation;

Authors response: We thank the Reviewer for the comment. Sample size calculation was provided in the materials and methods section.

Did you randomly distributed the teeth between the groups?

            Authors response: Yes, we did. It is described at the beginning of the material and methods section, 2.1 Study design: ‘‘Specimens were randomly (https:// randomizer.org) distributed among 6 study groups (n=14)…’’

How was the finishing procedure? Glazing? Polishing? It was the same between both materials? 

           Authors response: Restorations were polished. It is described in the text as ‘‘All restorations were finally polished with rotation silicon points and brushes.’’

Information about surface etching is missing. PICN and LD present different silica content and therefore should be properly etched. Please clarify.

            Authors response: We thank the Reviewer for the comment. It was clarified in the text.

Why 50 N in the chewing load simulator? This is a very low load for posterior tooth.

         Authors response: 50 N was decided in accordance with previously published articles to simulate in-vitro as close as possible to an in-vivo environment. See:

- De Kuijper, M.; Gresnigt, M.; van den Houten, M.; Haumahu, D.; Schepke, U.; Cune, M.S. Fracture Strength of Various Types 
of Large Direct Composite and Indirect Glass Ceramic Restorations. Oper. Dent. 2019, 44, 433–442. 
                                                                                                                                                                                    - Baldi A, Comba A, Michelotto Tempesta R, Carossa M, Pereira GKR, Valandro LF, Paolone G, Vichi A, Goracci C, Scotti N. External Marginal Gap Variation and Residual Fracture Resistance of Composite and Lithium-Silicate CAD/CAM Overlays after Cyclic Fatigue over Endodontically-Treated Molars. Polymers (Basel). 2021 Sep 4;13(17):3002. doi: 10.3390/polym13173002.                                                                                          - Baldi A, Comba A, Ferrero G, Italia E, Michelotto Tempesta R, Paolone G, Mazzoni A, Breschi L, Scotti N. External gap progression after cyclic fatigue of adhesive overlays and crowns made with high translucency zirconia or lithium silicate. J Esthet Restor Dent. 2022 Apr;34(3):557-564. doi: 10.1111/jerd.12837.

How the “events were recorded” during fatigue? In steps? After a specific number of cycles? Automatically by the fatigue machine software?

               Authors response: We thank the reviewer for the comment.  The events were automatically recorded by the fatigue machine software.

What was the aim of topic 2.8? It is not clear.

               Authors response: SEM analysis was performed to visually show different surface characteristics of the tested materials related to different preparation designs.  It was rephrased in the text to make it more clear.   The intent was to show readers how different materials behave at the interface with tooth structure before and after fatigue.  

The magnification described in the text and in the figures are different.

               Authors response: The Authors thank the Reviewer for pointing out the mistake. Magnification was corrected in the text. ‘‘The finished replicas were placed on aluminum stubs, gold was sputter-coated on them, and they were then analyzed under SEM (Phenom, FEI, Amsterdam, The Netherlands) at a magnification of 500x. The marginal quality of interfaces was assessed using SEM analysis..’’

Provide the normality test to justify your parametric statistic.

Authors response: we thank the Reviewer for the comment. Normality test was added in the statistical analysis section.

Describe the Tukey test in table 3.

            Authors response: It was added.

SEM figures are not clear. What should the reader see there? what region of the restoration are those?

           Authors response:  SEM images provided show the adhesive interface between the restoration and tooth. The intent was to show readers how different materials behave at the interface with tooth structure before and after fatigue.

Discussion:

Discussion should be improved with polymerization shrinkage stress between different preparation. Check the reference previously suggested and improve your text.

                Authors response: We thank the reviewer for the comment. The topic was added in the discussion.

What is the effect of different elastic modulus in the set?

               Authors response: We thank the Reviewer for the comment. The Authors apologize for not answering this point,  but the point is not clear for them and consequently it was not possible to answer it. Is it possible to reformulate it?

Discuss how handle both materials are, since one you need to crystalize before its use and the other is already fully-finished in the block.

                Authors response: We thank the Reviewer for the comment. It was added in the discussion.

How the simulation or not of periodontal ligament could affect your results?

                 Authors response: We thank the Reviewer for the comment. Periodontal ligament was simulated. It was added in the materials and methods section.

Round 2

Reviewer 3 Report

The authors performed all requested revisions. The text has been improved and the article can be considered accepted for publication. Congratulations for the excellent work.